

# From Field to Phone: A Karst Camp Chronicle

Rachel Bosch[1]

[1]Department of Geology, University of Cincinnati, Cincinnati, Ohio, 45150, U.S.

*Correspondence to*: Rachel Bosch (boschrf@mail.uc.edu)

**Abstract.** During the summer of 2020, many geology field camps were cancelled due to the COVID-19 pandemic, including the Karst Geomorphology field course I was scheduled to co-teach through Western Kentucky University. When the National Association of Geoscience Teachers (NAGT) in collaboration with the International Association for Geoscience Diversity (IAGD) began the project of supporting working groups to create online field experience teaching material, I saw an opportunity. From my field camp syllabus, I created two activities that are now freely available as peer-reviewed Exemplary Teaching Activities on the Science Education Resource Center (SERC) Online Field Experiences repository: "Karst Hydrogeology: A virtual field introduction using © Google Earth and GIS" and "Karst Hydrogeology and Geomorphology: A virtual field experience using © Google Earth, GIS, and TAK [Topographic Analysis Kit]." Each product includes a student handout, an instructor workflow reference, a grading, and NAGT-established learning objectives. The introductory activity is the more basic of the two, is expected to take about one 8-hour day to teach, and walks students through all the steps, as well as providing global examples of karst landscapes to virtually explore. The other activity, Karst Hydrogeology and Geomorphology, assumes student familiarity with © Google Earth, GIS, and karst drainage systems, and is expected to take about twice as long as the introductory activity to teach. To make these learning opportunities financially accessible, all software required for the activities is open-source and alternative workflows for the introductory module are provided so that the entire exercise can be completed using a smartphone. In addition to providing online capstone activities in the time of a pandemic, these activities provide alternative learning experiences to traditional field camps that are inclusive for all geoscience students. In my home department of the University of Cincinnati, I had been contacted by students needing to find capstone experiences when their field camps were cancelled. Responding to this need and providing a virtual alternative for years to come, I reviewed all SERC activities that had been generated during the NAGT/IAGD joint effort. I selected a subset of those to assemble into three learning tracks, each one providing learning hours equivalent to a traditional field camp, that have been added to the course offerings at the University of Cincinnati Department of Geology.

## 1 Introduction

This paper presents in detail two virtual field experiences that I designed, Karst Hydrology: A Virtual Field Introduction Using © Google Earth and GIS, and Karst Hydrogeology and Geomorphology: A virtual field experience using © Google Earth, GIS, and TAK. I then describe how I incorporated those activities, along with several others from the Science Education Resource Center (SERC) Online Field Experiences repository



(https://serc.carleton.edu/NAGTWorkshops/online_field/activities.html), to create Virtual Capstone Pathways for undergraduate geology majors in the Department of Geology at the University of Cincinnati.

Both activities described in this paper are now available in the SERC repository as part of the **Teaching with Online Field Experiences Exemplary Collection**. This exemplary rating is an official designation resulting from the peer review process (Burmeister et al., 2020) for the Teach the Earth portal for Earth Education.

These activities were designed with a constructivist teaching approach in mind (Brooks and Brooks, 1999). In each activity, students are guided through the technical details of using opensource software tools to select and explore a karst landscape so that they may construct knowledge about the hydrology and geomorphology of that region. While both activities require Internet access for participation, I considered nonphysical barriers such as limited income or restricted access to other resources. Therefore, all parts of both activities were built around using opensource software. Additionally, if a learner does not have a laptop or other person computer in their home, I designed an alternative workflow for Karst Hydrology: A Virtual Field Introduction Using © Google Earth and GIS such that the entire activity can be completed using a smartphone. This approach, considering universal design for learning (UDL), provides students with multiple means of representation for an accessible learning experience. Additionally, the learning assessment for these activities is written such that there is flexibility in the way individual students or small groups of students present their findings so that multiple means of expression are built in. These activities are designed for 100% online delivery, either synchronously, asynchronously, or using a combination of those approaches. Addressing physical and nonphysical barriers, incorporating UDL, and teaching field experiences synchronously online, have been proven to provide more inclusive learning experiences (Carabajal et al., 2017).

## 1.1 Learning Objectives

The group guiding the NAGT/IAGD joint effort to develop remote learning experiences for the 2020 field camp season collaborated to determine a set of learning objectives (NAGT, 2020). All activities developed as a part of this initiative were expected to address as many of these as possible:

1. Design a field strategy to collect or select data in order to answer a geologic question.
2. Collect accurate and sufficient data on field relationships and record these using disciplinary conventions (field notes, map symbols, etc.).
3. Synthesize geologic data and integrate with core concepts and skills into a cohesive spatial and temporal scientific interpretation.
4. Interpret earth systems and past/current/future processes using multiple lines of spatially distributed evidence.
5. Develop an argument that is consistent with available evidence and uncertainty.
6. Communicate clearly using written, verbal, and/or visual media (e.g., maps, cross-sections, reports) with discipline-specific terminology appropriate to your audience.





7.  Work effectively independently and collaboratively (e.g., commitment, reliability, leadership, open for advice, channels of communication, supportive, inclusive).

8.  Reflect on personal strengths and challenges (e.g., in study design, safety, time management, independent and collaborative work).

9.  Demonstrate behaviors expected of professional geoscientists (e.g., time management, work preparation, collegiality, health and safety, ethics).

## 70  2 Karst Hydrology: A Virtual Field Introduction Using © Google Earth and GIS

Through the activity, Karst Hydrology: A Virtual Field Introduction Using © Google Earth and GIS, https://serc.carleton.edu/NAGTWorkshops/online_field/activities/237039.html, students have the opportunity to select and virtually explore the hydrogeology and geomorphology of a karst landscape using © Google Earth, lidar data-sourced digital elevation models (DEM) and geologic maps, and GIS software (QGIS) such that they gain an understanding of karst

landscapes and their associated hazards, access to and analysis of internet-based remote sensing data, and verbal and written communication of scientific information. This basic activity is suitable for use in upper-level undergraduate geomorphology or groundwater hydrogeology courses, or as part of a capstone activity for graduating seniors. The main concepts explored are karst geomorphology, karst hydrogeology, © Google Earth image interpretation, and basic GIS landscape analysis.

### 2.1 Expectations

While there is no specific prerequisite coursework, students are expected to have familiarity with the concept of karst landscapes and topographic map reading. Additionally, previous experience with © Google Earth and other geographical information systems is helpful, but not necessary. This introductory activity is expected to take about half of a day to complete and can be taught as a stand-alone exercise or in conjunction with other modules to build a capstone field experience.

### 85  2.2 Learning Goals

The learning goals for this introductory karst activity are consistent with the NAGT/IAGD effort objectives outlined in the Introduction:

- Content/concepts goals for this activity
  - o  Visually identify karst landscapes (particularly in contrast to fluvial landscapes) from aerial imagery.
- o  Interpret topographic maps to determine drainage patterns.
- Higher order thinking skills goals for this activity





- o Compare and contrast the water drainage patterns of a karst watershed with a surface stream network or porous-media groundwater aquifer.
  - o Integrate digital and analog data.
  - o Analyze digital and analog data to draw conclusions about landscape-associated hazards.
- Other goals for this activity
  - o Navigate © Google Earth.
  - o Search the Internet, including USGS's Earth Explorer website.
  - o Manipulate data in a GIS for analysis and presentation.
  - o oral presentation or video presentation
  - o teamwork synchronously and asynchronously
  - o technical writing
  - o reflection
  - o self-assessment
  - o data management
  - o independence
  - o personal management
  - o time management
  - o leadership

## 110   2.3 Activity description and teaching materials

More than 25 percent of the world's population either lives on, or obtains their water from, karst landscapes (Maupin and Barber, 2005). It is therefore important that we understand the drainage patterns, potential hazards to humans, and potential threats to water quality that are unique to karst landscapes. In this exercise, students select and virtually explore a karst landscape. Materials available on the SERC repository include a student handout, teaching notes, a grading rubric, and
several Internet links for background and supplemental information. The teaching notes include a step-by-step walkthrough of the activity procedure, including screenshots of the anticipated outcomes. This way the modules can be available to a wide range of teachers and learners.

For this activity, students need access to an Internet enabled laptop or other device; © Google Earth on web, mobile, or desktop; and a geographic information system (GIS). QGIS (https://www.qgis.org/en/site/) is recommended as a free and
open source option which works best as a desktop download. As part of this exercise, I want students to experience manipulating DEM data with a GIS. However, I realize that not all students have access to a laptop, and GIS tools in mobile devices may not offer full functionality. One option for a fully mobile-based activity is to bypass the GIS steps and move straight from © Google Earth to accessing a pre-existing topographic map. Mobile apps change frequently, so there is a legacy issue in promoting specific apps. However, USGS has created a great tutorial video (closed-captioning embedded in





the video) for how to access topo maps from a mobile device: https://www.usgs.gov/media/videos/using-us-topo-and-historic-topo-maps-your-mobile-device. For this approach, students would skip steps 2, 3, and 4 (which are outlined below), following the procedure in the USGS video. They can then use a drawing app on their device to complete step 5.

Students may work in groups or independently to complete the activity. Presentations may be delivered as a group, and final reports should be written and submitted independently.

▪ The activity begins by providing resources for students to review the basics of karst hydrology. On the student handout, there are © Google Earth links for seven different field areas from around the world:

- Central Kentucky Karst, USA
- El Sotano de las Golondrinas, Mexico
- Caverna de Santana, Brazil

- Sof Omar Cave, Ethiopia
- Postojna Cave, Slovenia
- Tenglong Cave, China
- Waitomo Cave, New Zealand

Learners can browse and select among these options for a karst region to focus on for the remainder of the exercise (Fig.1).

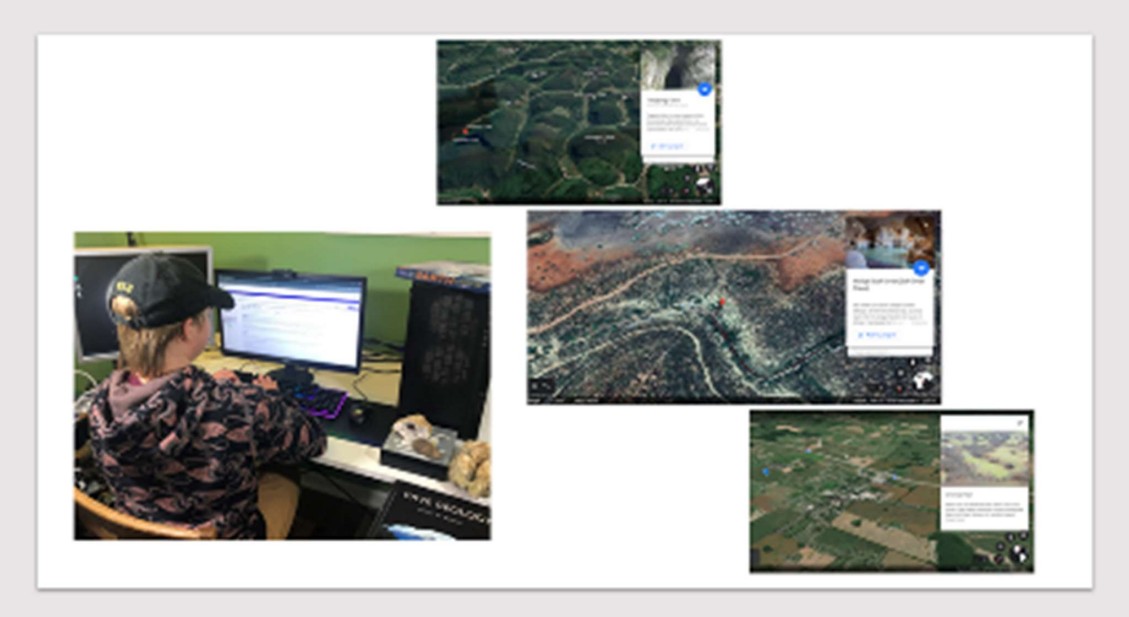


**Figure 1. Student choosing a karst area to study. Images of the © Google Earth pages for Tenglong Cave, China, Sof Omar Cave, Ethiopia, and the Central Kentucky Sinkhole Plain, United States, as examples. Photo by author.**



1. Once students have selected their karst landscape, they need to acquire topographic and geologic map information. For locations in the United States, Earth Explorer (https://earthexplorer.usgs.gov/) is a good source for SRTM digital elevation model (DEM) files. Students who choose sites outside of the United States can still find their DEM data, but will need to do some internet searching to obtain it. A walk-through with screenshots of this workflow is available in the Teaching Notes.

2. The DEM file then needs to be uploaded to a GIS. For many DEMs, students will need to find the appropriate coordinate reference system (CRS) and reproject their raster. References are provided for students to review the Universal Transverse Mercator System and find UTM zones, along with detailed directions for reprojecting DEMs in QGIS.

3. After their project is in the correct CRS, they generate a Hillshade layer to better visualize the topography (Fig. 2). They then answer the following questions: What karst aquifer region did you select? What UTM Zone is this field site in? What color band worked best for your visualization of the topography? What does the Hillshade function do? How is it helpful?

4. This projected and shaded map should now render a more accurate and realistic visualization of the chosen field site. To better understand the drainage patterns of this landscape, they extract two sets of topographic contour lines: one set for all the contour lines and a second, more widely spaced set, for index contours. They are asked what contour intervals they chose and why (Fig. 3).

5. Now that students have detailed topographic maps with contour intervals, they are given a resource to revisit the rule of V's for determining flow paths over land surfaces (Olivas, 2017). If students have access to a printer, they can print out a paper copy of the map they built and draw the drainage patterns in with a pencil. There are two digital options for drawing in the water flow paths. For the first, students can export the image of their map in QGIS as png format and then use a photo editor to draw flow paths on their maps. There are many software packages available to complete this step. A couple of opensource options are © Google slides (Fig. 4) and Inkscape. Students with more GIS experience may want to work directly in the GIS and make new vector layers to create their surface flow paths. Here are the questions posed to students after this step: Describe the flow paths you drew on your map. What challenges or obstacles did you encounter while determining the routes water would take? What environmental or natural-disaster hazards do you think might be issues in this landscape?





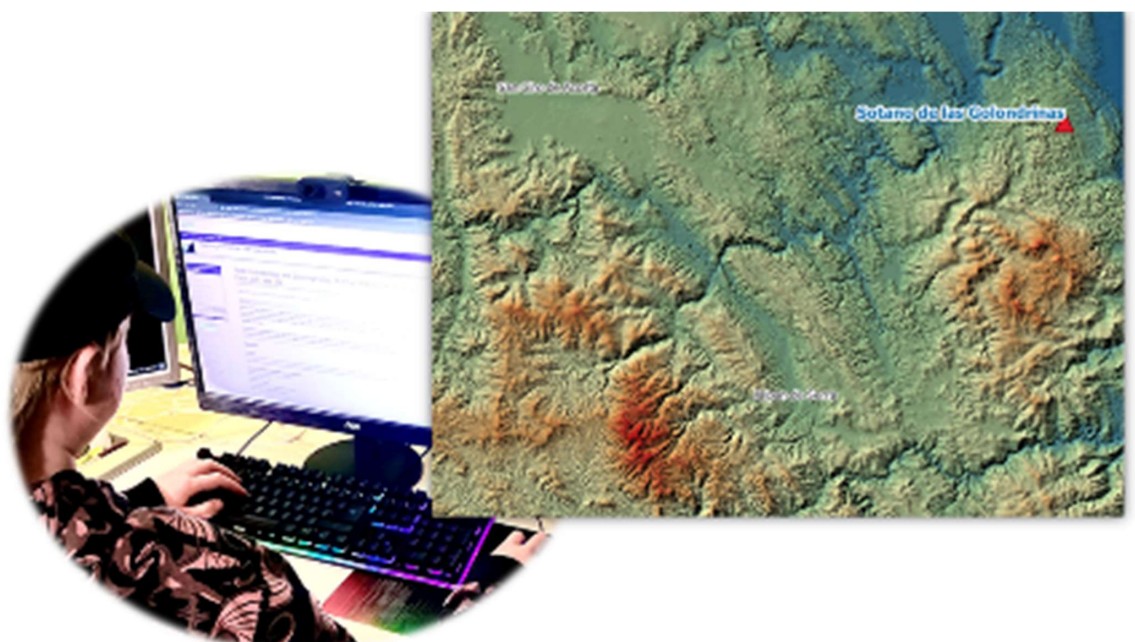

**Figure 2. Reprojected, hillshaded, and colored-by-elevation digital elevation model of the Sotano del las Golondrinas area, Mexico, in QGIS. Photo by author.**



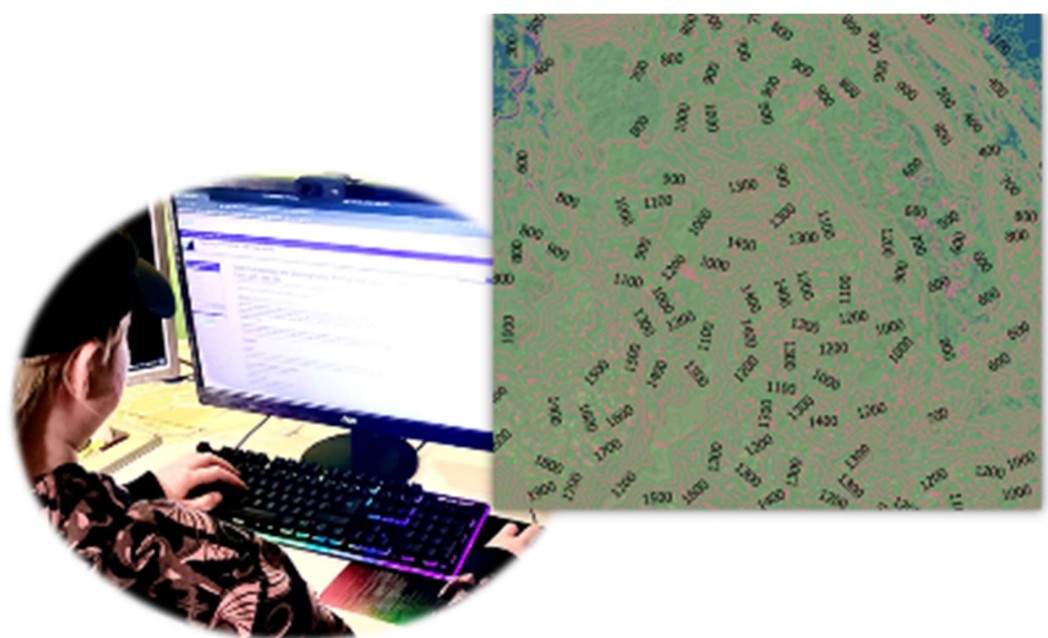


**Figure 3. El Sotano de las Golondrinas area with elevation contour layers. Photo by author.**

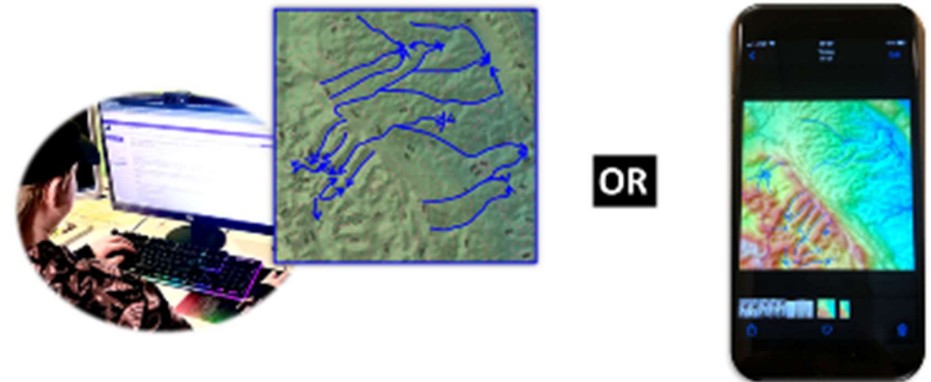

**Figure 4. Manually drawn water flow paths created in © Google Slides (left) and iPhone photo markup (right). Photo by author.**





**2.4 Assessment**

Questions for students are provided in the student handout (Appendix A). These are to guide students' thinking as they work through the activity. They can then use their responses to the questions as they prepare their presentations and write their final reports.

After completing the exercise as individuals or in small groups, students share their findings with the whole class. This can

happen virtually or in person as circumstances dictate. Each presentation can happen as a slideshow or as a video made by the student(s), or in any way that works for instructor(s) and student(s).

Each student also writes a formally structured report (Title, author's name, date, abstract, introduction, methods, results, discussion, conclusion). Within the report or as a separate document, they should reflect on their experience with this activity and assess their level of understanding before and after the activity of a.) © Google Earth, b.) GIS, c.) UTM CRS, d.)

topographic map interpretation, and e.) karst hydrogeology. A rubric to guide the grading of these materials is provided with the activity on the SERC repository (https://d32ogoqmya1dw8.cloudfront.net/files/teachearth/activities/grading_rubric_karst_intro_activity.docx).

**3 Karst Hydrogeology and Geomorphology: A virtual field experience using © Google Earth, GIS, and TAK**

While completing Karst Hydrology and Geomorphology: A virtual field experience using © Google Earth, GIS, and TAK,

https://serc.carleton.edu/NAGTWorkshops/online_field/activities/237267.html, students have the opportunity to select and virtually explore the hydrogeology and geomorphology of a karst landscape using © Google Earth, lidar data-sourced DEM(s), geologic maps, GIS software, and topographic analysis software packages such that they gain an understanding of karst landscapes and their associated hazard risks, access to and analysis of internet-based remote sensing data, design of field strategy, and verbal and written communication of scientific information. This activity incorporates and builds upon the

material covered in Karst Hydrogeology: A virtual field introduction using © Google Earth and GIS. This advanced activity, like the companion introductory activity, is suitable for use in upper-level undergraduate or graduate geomorphology or groundwater hydrogeology courses, or as part of a capstone activity for graduating seniors. The main concepts explored are karst geomorphology, karst hydrogeology, © Google Earth image interpretation, GIS landscape analysis, hypothesis development, and field strategy planning.

**3.1 Expectations**

This advanced module assumes that students have some prior experience in use of © Google Earth and other geographic information systems, as well as familiarity with the concept of karst landscapes, topographic map reading, and geologic map reading. This can be taught as a stand-alone exercise or in conjunction with other modules to build a capstone field experience and is expected to take one to two days to complete.



**3.2 Learning Goals**

The learning goals for the advanced virtual karst activity are also consistent with the NAGT virtual field learning objectives presented above:

- Content/concepts goals for this activity
    - Visual identification of karst landscapes (particularly in contrast to fluvial landscapes) from aerial imagery.
- Analog vs. digital topographic map interpretation to determine drainage patterns.
    - Effective field strategy planning to address an original hypothesis.
- Higher order thinking skills goals for this activity
    - Compare and contrast the ways karst drainage basins behave differently than purely surface stream or porous-media groundwater.
- Analyze digital and analog data to draw conclusions about landscape-associated hazards.
    - Compare and contrast analog with digitally automated analyses.
    - Formulate hypotheses using analog and digital data.
    - Develop an experimental strategy to test these hypotheses.
- Other skills goals for this activity
- Georeferencing analog data to a GIS
    - Geologic history construction
    - Navigating © Google Earth
    - searching the WWW and/or USGS's EarthExplorer website
    - manipulating data in a GIS for analysis and presentation
- oral presentation or video presentation
    - teamwork synchronously and asynchronously
    - technical writing
    - Reflection
    - Self-assessment
- data management
    - Independence
    - personal management
    - time management
    - leadership





**3.3 Activity description and teaching materials**

This advanced activity is similar to the introductory activity in that students select and virtually explore a karst landscape. Materials available on the SERC repository include a student handout, teaching notes, a grading rubric, and several Internet links for background and supplemental information. The teaching notes include a step-by-step walkthrough of the activity procedure, including screenshots of the anticipated outcomes. This way the modules can be available to a wide range of teachers and learners.

For this activity, students need access to an Internet enabled laptop. Prior to the activity, hey will need to download and install the following software packages: © Google Earth on web or desktop, a GIS (QGIS is a free and open source option), and Topographic Analysis Kit (TAK) (free, open source software package available at Github, https://github.com/amforte/Topographic-Analysis-Kit). Students may work in groups or independently to complete the activity and presentation. Final reports should be written and submitted independently.

This activity begins with students exploring the World Karst Aquifer Map (Goldscheider, 2021; Fig. 5) and its associated article by Goldscheider et al. (2020). Instead of having seven discrete choices as in the introductory activity, they are presented with the opportunity to use these resources to select any karst area to study.

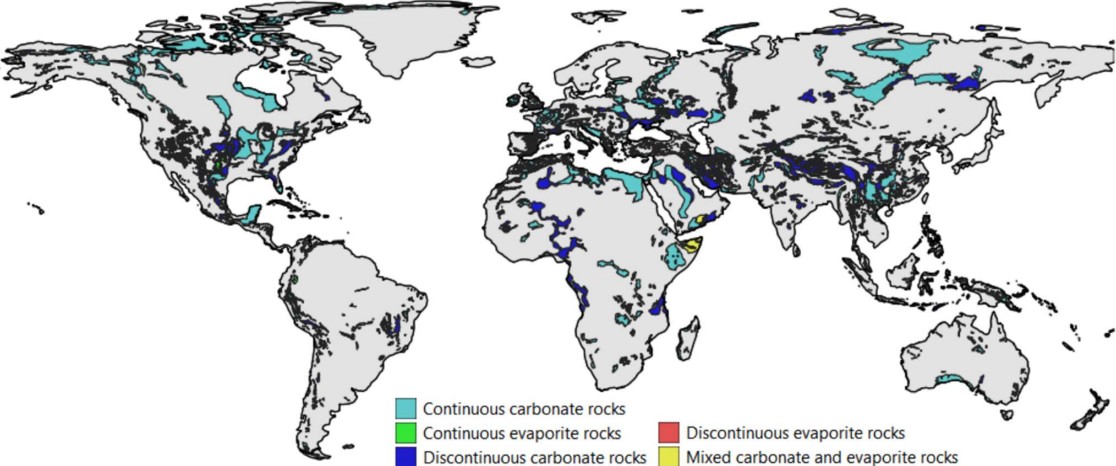

**Figure 5. World Karst Aquifer Map (cartography by author in QGIS, data from Goldscheider, 2021).**

1.  Once students have selected their karst landscape, they need to acquire topographic and geologic map information. For locations in the United States, Earth Explorer is a good source for digital elevation model (DEM) files. Students who choose sites outside of the United States can still find their DEM data, but will need to do some internet searching to obtain them. A walk-through with screenshots of this workflow is available in the Teaching Notes.

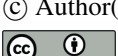



They will then follow the instructions for the introductory activity described above, following steps 2 through 5.

2.    To determine flow paths more objectively, students use TAK. This software uses a set of Matlab functions written by Adam Forte which build upon the functionality of TopoToolbox by Wolfgang Schwanghart and Dirk Scherler. They load the Reprojected DEM into TAK and run "MakeStreams" to perform automated flow routing. This is the only function used in the activity, but hopes are that some students' interest will be piqued to explore further landscape analysis.

3.    Students then add the streams vector layer generated by TAK to their GIS project.

4.    The final piece of data students add to their GIS is regional geology. The student handout provides instruction for importing a shapefile or for manually georeferencing an analog geologic map.

5.    They are then ready to begin the analysis portion of the activity. This analysis will be presented to the class if they are working in a cohort and the responses to the following steps will be written into their final report.

6.    There will be differences between the manual flow routing drawn by students and the flow paths modeled by the routing algorithms in TAK (Fig. 6). They are called upon to speculate about the source of those differences, and then to consider the strengths and weaknesses of each approach. They then decide which method they think provides better results. This conclusion may vary based upon the student or the selection of field study location.

7.    Using observations of the geology, the topography, and the hydrology, students construct a geologic/geomorphic history of their study area (Fig. 7). They are asked to determine the sequence of events at this site, and to consider in particular, depositional, tectonic, and erosional events. They are also asked how any stream network (or lack thereof) evolved.

8.    Now that they have constructed a story, they are reminded that they have done this with limited evidence. They are asked think about their level of confidence for each event and state which events they think may need more support. They build those into hypotheses.

9.    Additional hypothesis may be developed concerning environmental or natural hazards they think might be issues with this landscape.

10.    The final step is to describe a potential experiment that would test one or more of the hypotheses proposed in 8 and 9. The students first state what data they would need to collect to support or refute their hypotheses and speculate as to the kinds of results that may be obtained for different types of data as well as what implications these results might have for each hypothesis. Finally, they are asked what field, laboratory, or numerical techniques





would be required to obtain these data, and to be specific as if they were planning for field work, lab work, or numerical modeling.

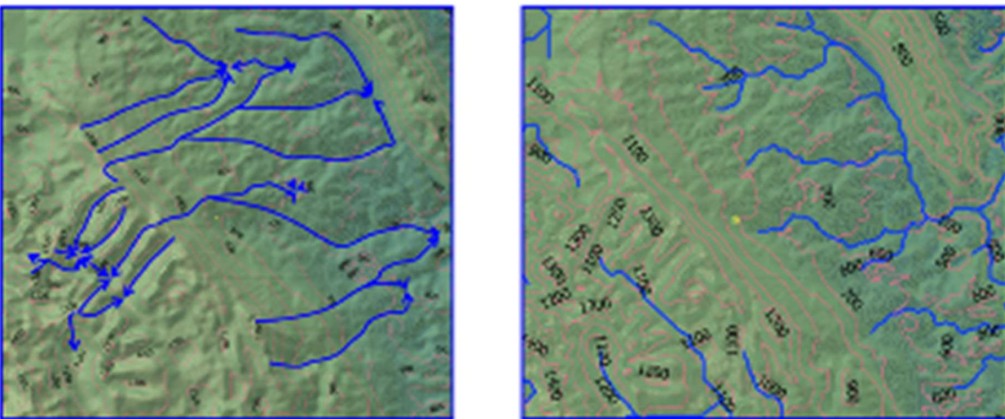

Compare manual and automatic flow routing results

**Figure 6. Manually drawn water flow routing (left) and automated flow routing performed in TAK (right).**



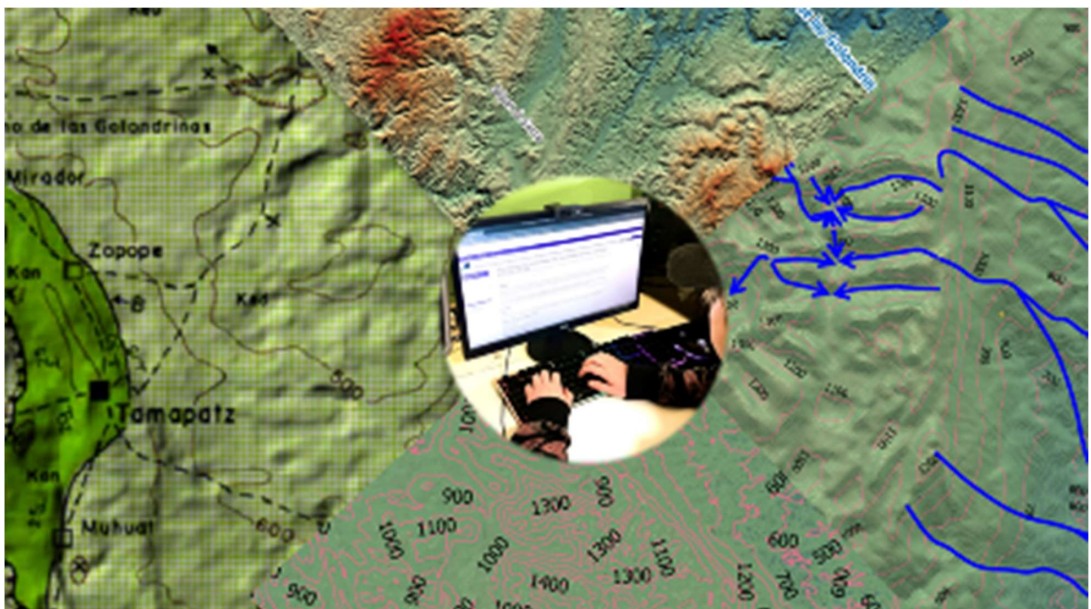

**Figure 7. Advanced activity synthesis, clockwise from left: analog geological map draped over topography using QGIS, DEM with hillshade in QGIS, manually drawn flow routing, and elevation contour layers in QGIS. Photo by author.**

### 3.4 Assessment

Questions for students are provided in the student handout (Appendix B). These are to guide students' thinking as they work through the activity. They can then use their responses to the questions as they prepare their presentations and write their

final reports.

After completing the exercise as individuals or as small groups, students share their findings with the whole class. This can happen virtually or in person as circumstances dictate. Each presentation can happen as a slideshow or as a video made by the student(s), or in any way that works for both instructor(s) and student(s).

Each student writes a formally structured report (title, author's name, date, abstract, introduction, methods, results,

discussion, conclusion). Within the report or as a separate document, they should reflect on their experience with this activity and assess their level of understanding before and after the activity of a.) © Google Earth, b.) GIS, c.) UTM CRS, d.) topographic map interpretation, e.) TAK, e.) karst hydrogeology, f.) geologic history construction, g.) hypothesis formulation, and h.) scientific experiment design.

A    rubric    to    guide    the    grading    of    these    materials    is    provided    with    the    activity    on    SERC

([https://d32ogoqmya1dw8.cloudfront.net/files/NAGTWorkshops/online_field/activities/grading_rubric_karst_hydrogeology](https://d32ogoqmya1dw8.cloudfront.net/files/NAGTWorkshops/online_field/activities/grading_rubric_karst_hydrogeology) _geomorphology_virtual_field_experience_us.docx).





## 4 Virtual Capstone Pathway Design

When I was contemplating how to disseminate the karst products described above and a variety of other geological products created by others during the push for virtual field learning, I was contacted by a student I had met in my capacity as a

teaching assistant for *Geomorphology* in Fall 2018. They asked whether there were any virtual capstone opportunities to help them meet their graduation requirements on time. This provided me with the motivation to organize some of these great activities into learning pathways that we could offer to students at the University of Cincinnati. I sorted existing modules from the SERC Teaching with Online Experiences page into three interest-based tracks: Planetary Geology, Traditional Geology, and Environmental Geology (Tables 1, 2, and 3, respectively). Since these activities include information on SERC

about how long they should take to complete, I was able to design each path such that they have an estimated instructional equivalent to 28 days in the field—approximating a four- to six-week field camp—and they would thus each be suitable for a senior capstone experience. This has been announced to the departmental community as a virtual capstone option available for students. In addition to providing a safer route for students to meet their graduation criteria during a pandemic, these Virtual Capstone Pathways are a robust and rigorous alternative for students who are not able to attend a traditional field

camp.

**Table 1. Modules for a Planetary Geology Virtual Capstone Pathway**

| Track: | Planetary Geology | |
|---|---|---|
| Module Name | Field day equivalent | URL |
| Uncertainty Modules | 0.5 | https://serc.carleton.edu/NAGTWorkshops/online_field/activities/237278.html |
| JMARS training | 0.5 | https://serc.carleton.edu/teachearth/activities/236955.html |
| Fold Analysis Challenge | 1 | https://serc.carleton.edu/geode/activities/217440.html |
| Using StraboSpot for Field Sedimentology & Stratigraphy | 1 | https://serc.carleton.edu/NAGTWorkshops/online_field/activities/237339.html |
| Introduction to Planetary Mapping | 1 | https://serc.carleton.edu/teachearth/activities/236956.html |
| Geologic Mapping of a Virtual Landscape | 2 | https://serc.carleton.edu/NAGTWorkshops/online_field/activities/236670.html |
| Orienteering in Minecraft | 1 | https://serc.carleton.edu/NAGTWorkshops/online_field/activities/237088.html |





| Karst Hydrogeology: A virtual field introduction using © Google Earth and GIS | 1 | https://serc.carleton.edu/NAGTWorkshops/online_field/activities/237039.html |
| Volcano mapping on Mount Cleveland volcano | 5 | http://master32.cas.usf.edu/fieldschool/bearing/bearing3.html http://master32.cas.usf.edu/fieldschool/UTM/utm1.html http://master32.cas.usf.edu/fieldschool/volume/volume.html http://master32.cas.usf.edu/fieldschool/cleveland-map/cleveland-map.html http://master32.cas.usf.edu/fieldschool/isomass/isomass.html |
| Virtual Geologic Mapping Exercise at Lough Fee, Ireland | 3 | http://csmgeo.csm.jmu.edu/Geollab/Whitmeyer/geode/Lough_Fee/ https://serc.carleton.edu/NAGTWorkshops/online_field/activities/237160.html |
| Geologic Mapping of a Virtual Landscape II – Three River Hills | 3 | https://serc.carleton.edu/NAGTWorkshops/online_field/activities/237061.html |
| Geologic mapping on Mars | 4 | https://serc.carleton.edu/NAGTWorkshops/online_field/activities/237044.html |
| A virtual fieldtrip on the coastal geomorphology of Naxos Isl. | 5 | https://serc.carleton.edu/NAGTWorkshops/online_field/activities/238032.html |
| | 28 | |

**Table 2. Modules for Traditional Geology Virtual Capstone Pathway**

| Track: | Traditional Geology | |
|---|---|---|
| Module Name | Field day equivalent | URL |
| Uncertainty Modules | 0.5 | https://serc.carleton.edu/NAGTWorkshops/online_field/activities/237278.html |
| Fold Analysis Challenge | 1 | https://serc.carleton.edu/geode/activities/217440.html |



| | | |
|---|---|---|
| Using StraboSpot for Field Sedimentology & Stratigraphy | 1 | https://serc.carleton.edu/NAGTWorkshops/online_field/activities/237339.html |
| Reconnaissance stratigraphy and mapping of the Frying Pan Gulch, MT | 5 | https://serc.carleton.edu/NAGTWorkshops/online_field/activities/237089.html |
| Karst Hydrogeology: A virtual field introduction using © Google Earth and GIS | 1 | https://serc.carleton.edu/NAGTWorkshops/online_field/activities/237039.html |
| Volcano mapping on Mount Cleveland volcano | 5 | http://master32.cas.usf.edu/fieldschool/bearing/bearing3.html http://master32.cas.usf.edu/fieldschool/UTM/utm1.html http://master32.cas.usf.edu/fieldschool/volume/volume.html http://master32.cas.usf.edu/fieldschool/cleveland-map/cleveland-map.html http://master32.cas.usf.edu/fieldschool/isomass/isomass.html |
| Virtual Geologic Mapping Exercise at Lough Fee, Ireland | 3 | http://csmgeo.csm.jmu.edu/Geollab/Whitmeyer/geode/Lough_Fee/ https://serc.carleton.edu/NAGTWorkshops/online_field/activities/237160.html |
| Sage Hen flat | 6 | https://serc.carleton.edu/NAGTWorkshops/online_field/activities/238026.html |
| Remote Mapping and Analytical data integration: Coal Creek quartzite and Ralston shear zone, Colorado | 5 | https://serc.carleton.edu/NAGTWorkshops/online_field/activities/237694.html |

27.5

**Table 3. Modules for Environmental Geology Virtual Capstone Pathway**

| Track: | Environmental Geology | |
|---|---|---|
| Module Name | Field day equivalent | URL |



| | | |
|---|---|---|
| Uncertainty Modules | 0.5 | https://serc.carleton.edu/NAGTWorkshops/online_field/activities/237278.html |
| Fold Analysis Challenge | 1 | https://serc.carleton.edu/geode/activities/217440.html |
| Using StraboSpot for Field Sedimentology & Stratigraphy | 1 | https://serc.carleton.edu/NAGTWorkshops/online_field/activities/237339.html |
| Quaternary mapping in Bells Canyon as an introduction to ArcGIS Pro | 1 | https://serc.carleton.edu/NAGTWorkshops/online_field/activities/237280.html |
| Go with the Flow: A Virtual Field Experience on Groundwater Flow | 3 | https://serc.carleton.edu/NAGTWorkshops/online_field/activities/237415.html |
| Volcano mapping on Mount Cleveland volcano | 5 | http://master32.cas.usf.edu/fieldschool/bearing/bearing3.html<br>http://master32.cas.usf.edu/fieldschool/UTM/utm1.html<br>http://master32.cas.usf.edu/fieldschool/volume/volume.html<br>http://master32.cas.usf.edu/fieldschool/cleveland-map/cleveland-map.html<br>http://master32.cas.usf.edu/fieldschool/isomass/isomass.html |
| Virtual Geologic Mapping Exercise at Lough Fee, Ireland | 3 | http://csmgeo.csm.jmu.edu/Geollab/Whitmeyer/geode/Lough_Fee/<br>https://serc.carleton.edu/NAGTWorkshops/online_field/activities/237160.html |
| Birth of a River in Yellowstone National Park | 2 | https://serc.carleton.edu/NAGTWorkshops/online_field/activities/237410.html |
| Landslide Mapping and Analysis Module | 5 | https://serc.carleton.edu/NAGTWorkshops/online_field/activities/237687.html |
| Karst Hydrogeology and Geomorphology: A virtual field experience using ⓒ Google Earth, GIS, and TAK | 2 | https://serc.carleton.edu/NAGTWorkshops/online_field/activities/237267.html |
| A virtual fieldtrip on the coastal | 5 | https://serc.carleton.edu/NAGTWorkshops/online_f |



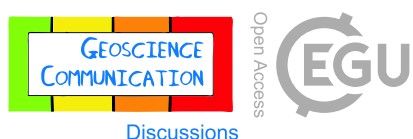

geomorphology of Naxos Isl.                    ield/activities/238032.html

28.5



**5 Virtual Capstone Pathways Implementation**

The Department of Geology at the University of Cincinnati first offered the Virtual Capstone Pathways during the Fall 2020 semester. Two students began work on Virtual Capstone Pathways tracks (Dietsch, 2020). This option is available for Spring
2021 in the course catalog as "Geology Capstone, 100% Online" and will be offered during Summer 2021, as well (Smilek, 2020).

**6 Sharing resources to the broader community**

In an effort to make these resources as widely available as possible, I presented the materials in this chapter at the Geological Society of America 2020 Connects Online meeting as a pre-recorded video, From Field to Phone: A Karst Camp Chronicle
(https://www.youtube.com/watch?v=3YvbUOlBRs0), with embedded English subtitles. The next step to broaden the reach of these materials was to localize the video, translating those subtitles into Spanish, Simplified Chinese, Japanese, German, and French (Fig. 8). The translations were performed using © Google Translate. For each subtitle in Chinese and Japanese, and for those where I was unsure about their accuracy in the other three languages, I copied the translated text and ran it back through © Google Translate to return it to English. I iterated through this process until the translation correctly retained the
intended meaning. A document was developed to explain this project in those six languages, containing 1) a link to the Teaching with Online Resources page, 2) links to both karst virtual field experience activities, 3) a link to a © Google doc with the virtual capstone pathways information, 4) a link to the GSA video on YouTube, and 5) the GSA presentation abstract (Appendix C).

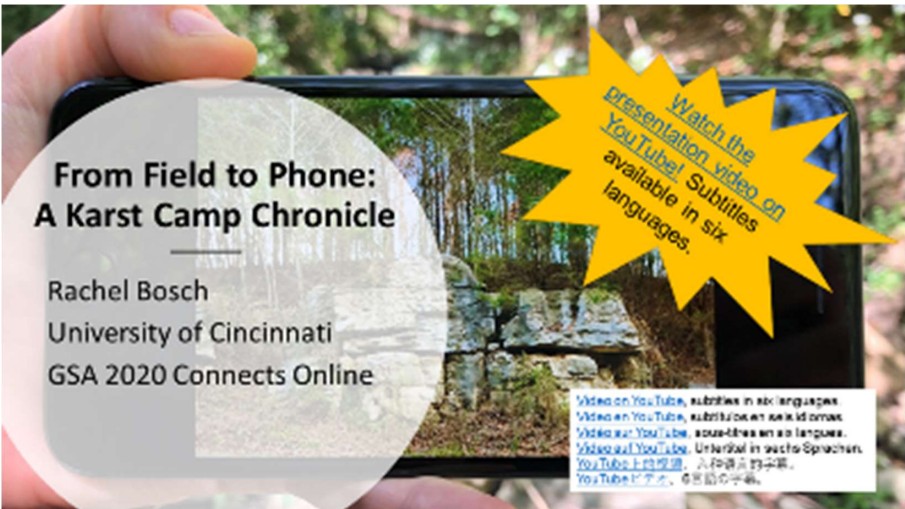

**Figure 8. Marketing image for virtual field experience activities GSA 2020 presentation YouTube video.**



## 7 Discussion

Mixed in with the challenges presented by the COVID-19 pandemic, there were many opportunities. Our communities have grown stronger and developed new strategies because we were forced to think, learn, and work in new ways during 2020. Remote learning and online conferences provided a chance to develop new tools that will make for more equitable academic

experiences in the years ahead. In addition to keeping everyone safe during the pandemic, the geoscience community used this time to perform some large-scale, collective introspection. New tools and ways of working, including the virtual field experiences and virtual capstone pathways described in this paper, are an instrumental part of our community working to become for welcoming, accessible, and inclusive.

## Acknowledgments

This work was made possible by Chris Atchison, Kurt Burmeister, Anne Egger, Katherine Ryker, and Basil Tikoff, who coordinated the geoscience community to join together during the pandemic of 2020 to provide a large set of freely available virtual field learning experience activities. Aida Farough and Bridget Mulvey, as co-leaders for the Hydrogeology/Environmental Geology working group provided invaluable support as I, and other members of the team, designed our activities. Thank you to Craig Dietsch and Krista Smilek in the UC Department of Geology for embracing my

Virtual Capstone Pathways design and incorporating it into the Department. Aaron Bird did wonderfully as cinematographer for the activity promotion video. I am grateful to Ian Castro, Brooke Crowley, and Chris Atchison for reviewing this paper prior to submission. Finally, thanks go out to Samuel Bosch-Bird for modeling as the "student" in all photos.

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

**Appendix A. Introductory Activity: Student Handout**

**Karst Hydrogeology: A virtual field introduction using © Google Earth and GIS**

o **By Rachel Bosch, University of Cincinnati**            karstgeomorph@gmail.com

**A.1 Summary**

Students will have the opportunity to select and virtually explore the hydrogeology and geomorphology of a karst landscape
using © Google Earth, lidar data-sourced DEM(s), and GIS software (QGIS) such that they gain an understanding of karst
landscapes and their associated hazards, access to and analysis of internet-based remote sensing data, and verbal and written
communication of scientific information.

**A.2 Activity Description**

Karst aquifers supply drinking water to 25% of our world's population. It is therefore important that we understand the
drainage patterns, potential hazards to humans, and potential threats to water quality that are unique to karst.

Prior to beginning this activity, download and install the following software packages: © Google Earth on web or desktop
(https://www.google.com/earth/versions/) and a GIS (QGIS is a free and open source option: https://www.qgis.org/en/site/).





1. Background. Review background information on karst and on the source of the digital elevation model (DEM) data used in this activity.

Background information on karst: https://link.springer.com/article/10.1007/s10040-016-1519-3, https://kgs.uky.edu/kgsweb/olops/pub/kgs/ic04_12.pdf, https://en.wikipedia.org/wiki/Karst, http://www.igme.es/boletin/2016/127_1/BG_127-1_Art-9.pdf.

Background on specific karst areas you can explore during this activity:

- Central Kentucky Karst, USA https://www.usgs.gov/science-support/osqi/yes/national-parks/mammoth-cave-
455       national-park,

http://www.igme.es/boletin/2016/127_1/BG_127-1_Art-9.pdf

- El Sotano de las Golondrinas, Mexico https://en.wikipedia.org/wiki/Cave_of_Swallows
- Caverna de Santana, Brazil https://en.wikipedia.org/wiki/Caverna_Santana
- Sof Omar Cave, Ethiopia

https://en.wikipedia.org/wiki/Sof_Omar_Caves

- Postojna Cave, Slovenia

https://www.postojnska-jama.eu/en/, https://www.slovenia.info/en/stories/karst, https://izrk.zrc-sazu.si/en/predstavitev#v

- Tenglong Cave, China

https://en.wikipedia.org/wiki/Tenglong_Cave

- Waitomo Cave, New Zealand

https://www.newzealand.com/us/waitomo-caves/, https://en.wikipedia.org/wiki/Waitomo_Glowworm_Caves

Background on the Shuttle Radar Topography Mission (SRTM) to acquire the data used in the DEMs recommended in this activity: https://www2.jpl.nasa.gov/srtm/

For an overview of karst aquifers on Earth, refer to the World Karst Aquifer Map (WOKAM), available at

https://www.whymap.org/whymap/EN/Maps_Data/Wokam/wokam_node_en.html. Use WOKAM to select an area of interest, browse © Google Earth to search for karst landforms, or use one of the following links to go directly to a karst area:

- Central Kentucky Karst, USA https://earth.google.com/web/search/smiths+Grove,+KY,+USA/@37.050087,-86.20808355,191.9936422a,4259.47719636d,35y,0h,0t,0r/data=CigiJgokCUJ0d1jEiEJAEea5ieZmhUJAGblW8S1QglXAIfK61_HlhlXA

- El           Sotano           de           las           Golondrinas,           Mexico https://earth.google.com/web/search/S%c3%b3tano+De+Las+Golondrinas,+Guadalupe,+Tamapatz,+San+Luis+Potosi,+Mexico/@21.5998365,-99.0989649,851.98715572a,974.07941593d,35y,0h,45t,0r/data=CrUBGooBEoMBCiUweDg1ZDY4NzUzMGM5ODE4Y2I6MHhiMTBkYjMzYzJlMmI4ZmY2GXVyhuKOmTVAIew84HBVxljAKkhTw7N0YW5vIERlIExhcyBH




- Caverna de Santana, Brazil https://earth.google.com/web/search/Caverna+de+Santana+-+SP-165,+Iporanga+-+State+of+S%c3%a3o+Paulo,+Brazil/@-24.5350913,-48.7020552,293.73887977a,950.74006882d,35y,0h,45t,0r/data=Cq8BGoQBEn4KJTB4OTRjNGEwMTVjN2I5Y2

- Sof Omar Cave, Ethiopia https://earth.google.com/web/search/Holqa+Soof+Umar(Sof+Omar+Cave),+Ethiopia/@6.9066847,40.8452162,122

- Postojna Cave, Slovenia
https://earth.google.com/web/search/Postojna+Cave,+Jamska+cesta,+Postojna,+Slovenia/@45.7830298,14.203841

- Tenglong Cave, China
https://earth.google.com/web/search/tenglong+cave/@30.331922,108.983322,1420.20643916a,896.95198141d,35y,

- Waitomo Cave, New Zealand https://earth.google.com/web/search/waitomo+cave/@-38.23941234,175.06756931,225.98846243a,29217.75304527d,35y,0h,0t,0r/data=CnYaTBJGCiQweDZkNmNiN2I

2. Data acquisition. Acquire topographic information for your chosen karst landscape. For locations in the United States, Earth Explorer is a good source for **SRTM DEM files** (https://earthexplorer.usgs.gov/). For sites outside of the US you can still find DEM data, but may need to do additional internet searching to obtain it.

3. Data processing.



a.  The DEM file then needs to be uploaded to a **GIS**. Check the properties of your DEM raster layer to see what coordinate reference system (CRS) it loaded in. For many DEMs, you will need to find the appropriate CRS and **reproject** the raster. For a review of the Universal Transverse Mercator (UTM) System, here is a link to the USGS fact sheet ([https://pubs.usgs.gov/fs/2001/0077/report.pdf](https://pubs.usgs.gov/fs/2001/0077/report.pdf)) and a world map of UTM zones ([https://maptools.com/tutorials/grid_zone_details](https://maptools.com/tutorials/grid_zone_details)). Another option is to use an interactive online map ([https://mangomap.com/robertyoung/maps/69585/what-utm-zone-am-i-in-](https://mangomap.com/robertyoung/maps/69585/what-utm-zone-am-i-in-)) to help determine the coordinate system for your location. The reproject task is performed by selecting the layer for the DEM raster data. Then click on the "raster" drop down menu. Go to "projections," and select "Warp (reproject)..." Then select a complete path for output and give a name to the output file for the reprojected map data.

b.  After the project is in the correct CRS, you can then choose a color scheme (right click on the layer > "properties" > "style" > "render type" > "singleband pseudocolor" > "generate a new color map" > select the desired color band > "classify") and make a **Hillshade** layer to better visualize the topography. To generate a Hillshade layer, again use the "raster" menu. Go to "Terrain analysis" > "Hillshade…"

   i.  *Questions: What karst aquifer region did you select? What UTM Zone is this field site in? What colorband worked best for your visualization of the topography? What does the Hillshade function do? How is it helpful?*

c.  To better understand the drainage patterns of this landscape, extract a set of topographic contour lines. Again use the "raster" menu. Go to "Extraction" > "Contour…" A good interval to start with a 20. If the contour lines end up looking too crowded or too spread out, you can make new contour layers with different intervals.

d.  Now that you have detailed topographic maps with contour intervals, you may want to revisit the rule of V's for determining flow paths over land surfaces ([http://uncivilengineer.net/2017/07/14/watercourses-and-ridges-on-topographic-maps-why-the-vs/](http://uncivilengineer.net/2017/07/14/watercourses-and-ridges-on-topographic-maps-why-the-vs/)).

If you have access to a printer, you can print out a paper copy of the map you built and draw the drainage patterns in with a pencil. There are two digital options for drawing in the water flow paths. For the first, you can export the image of their map in QGIS as png format. To do this go to the "Project" menu and select "Save as Image…" Then use a photo editor to draw flow paths on the map. If you have more GIS experience you may want to work directly in the GIS and make new vector layers to create surface flow paths.

   i.  *Questions: Describe the flow paths you drew on your map. What challenges or obstacles did you encounter while determining the routes water would take?*

Sharing science:

1.  After completing the exercise, as an individual or as part of a small group, present your findings to the whole class.



2.    Write a formally structured report (Title, author's name, date, abstract, introduction, methods, results, discussion, conclusion). Within the report or as a separate document, reflect on your experience with this activity and assess your level of understanding before and after the activity of a.) © Google Earth, b.) GIS, c.) UTM CRS, d.) topographic map

interpretation, and e.) karst hydrogeology.

**Appendix B. Advanced Activity: Student Handout**

**Karst Hydrogeology and Geomorphology: A virtual field experience using © Google Earth, GIS, and TAK**

o    **By Rachel Bosch, University of Cincinnati          karstgeomorph@gmail.com**

**B.1 Summary**

Students will have the opportunity to select and virtually explore the hydrogeology and geomorphology of a karst landscape using © Google Earth (or perhaps © Google Mars or © Google Moon if they so choose), lidar data-sourced DEM(s), geologic maps, GIS software, and topographic analysis software packages such that they gain an understanding of karst landscapes and their associated hazard risks, access to and analysis of internet-based remote sensing data, design of field strategy, and verbal and written communication of scientific information.

This activity incorporates and builds upon *Karst Hydrogeology: A virtual field introduction using © Google Earth and GIS.* If you have already completed the introductory activity, use your results from that activity and continue onto this activity with step 3.f.

**B.2 Activity Description**

Karst aquifers supply drinking water to 25% of our world's population. It is therefore important that we understand the

drainage patterns, potential hazards to humans, and potential threats to water quality that are unique to karst.

Prior to beginning this activity, download and install the following software packages: © Google Earth on web or desktop (https://www.google.com/earth/versions/); a GIS (QGIS is a free and open source option: https://www.qgis.org/en/site/); Topographic Analysis Kit (free, open source software package available at github: https://github.com/amforte/Topographic-Analysis-Kit)

1.    Background. Review background information on karst and on the source of the digital elevation model (DEM) data used in this activity.

Background information on karst: https://link.springer.com/article/10.1007/s10040-016-1519-3,

https://kgs.uky.edu/kgsweb/olops/pub/kgs/ic04_12.pdf, https://en.wikipedia.org/wiki/Karst,

http://www.igme.es/boletin/2016/127_1/BG_127-1_Art-9.pdf.

Background on the Shuttle Radar Topography Mission (SRTM) to acquire the data used in the DEMs recommended in this activity: https://www2.jpl.nasa.gov/srtm/



2. Data acquisition.

 a. For an overview of karst aquifers on Earth, refer to the **World Karst Aquifer Map (WOKAM)**, available at https://www.whymap.org/whymap/EN/Maps_Data/Wokam/wokam_node_en.html. Use WOKAM to select an area of interest or browse © Google Earth to search for karst landforms.

 b. As a base layer for GIS mapping of your karst area, load the WOKAM **shapefiles** to view the chosen area in the context of its broader karst region. WOKAM shapefiles can be found at https://produktcenter.bgr.de/terraCatalog/OpenSearch.do?search=473d851c-4694-4050-a37f-ee421170eca8&type=/Query/OpenSearch.do or in the attached zipped folder.

 c. Acquire topographic and geologic map information for your chosen karst landscape. For topography in the United States, Earth Explorer is a good source for **SRTM DEM files** (https://earthexplorer.usgs.gov/). For sites outside of the US you can still find DEM data, but may need to do additional internet searching to obtain it.

 d. Acquire geologic map data. Inside of the US, the **National Geologic Map Database** project should have what you need (https://ngmdb.usgs.gov/ngmdb/ngmdb_home.html). Outside of the US, it will vary country by country and more internet searching will be needed. Geologic information may be in a file format ready for import to a GIS or as a scanned image or pdf file.

3. Data processing.

 a. The DEM file then needs to be uploaded to a **GIS**. Check the properties of your DEM raster layer to see what coordinate reference system (CRS) it loaded in. For many DEMs, you will need to find the appropriate CRS and **reproject** the raster. For a review of the Universal Transverse Mercator (UTM) System, here is a link to the USGS fact sheet (https://pubs.usgs.gov/fs/2001/0077/report.pdf) and a world map of UTM zones (https://maptools.com/tutorials/grid_zone_details). Another option is to use an interactive online map (https://mangomap.com/robertyoung/maps/69585/what-utm-zone-am-i-in-) to help determine the coordinate system for your location. The reproject task is performed by selecting the layer for the DEM raster data. Then click on the "raster" drop down menu. Go to "projections," and select "Warp (reproject)..." Then select a complete path for output and give a name to the output file for the reprojected map data.

 b. After the project is in the correct CRS, you can then choose a color scheme (right click on the layer > "properties" > "style" > "render type" > "singleband pseudocolor" > "generate a new color map" > select the desired color band > "classify") and make a **Hillshade** layer to better visualize the topography. To generate a Hillshade layer, again use the "raster" menu. Go to "Terrain analysis" > "Hillshade…"

  i. *Questions: What karst aquifer region did you select? What UTM Zone is this field site in? What colorband worked best for your visualization of the topography? What does the Hillshade function do? How is it helpful?*





 c. The next layer to upload to GIS is geologic map information.

  i. If the geologic map data is in a proper file format for GIS, it will most likely need to be reprojected to the same CRS as the elevation data (see step 3a).

  ii. If the geologic map data is a scanned image, you have two options:


   1. Import the image to the GIS and Georeference it to align it with the map. The procedure for georeferencing analog images is covered in this webinar (https://www.youtube.com/watch?v=WbMdNvQcCOs); or

   2. Work side-by-side comparing the information from your geologic map with that on the GIS. This is less precise, but if you are careful you can make it work.


 d. To better understand the drainage patterns of this landscape, extract a set of topographic contour lines. Again use the "raster" menu. Go to "Extraction" > "Contour…" A good interval to start with a 20. If the contour lines end up looking too crowded or too spread out, you can make new contour layers with different intervals.

 e. Now that you have detailed topographic maps with contour intervals, you may want to revisit the rule of


  V's for determining flow paths over land surfaces (http://uncivilengineer.net/2017/07/14/watercourses-and-ridges-on-topographic-maps-why-the-vs/).

If you have access to a printer, you can print out a paper copy of the map you built and draw the drainage patterns in with a pencil. There are two digital options for drawing in the water flow paths. For the first, you can export the image of their map in QGIS as png format. To do this go to the "Project" menu and select "Save as Image…" Then use a photo editor to draw

630 flow paths on the map. If you have more GIS experience you may want to work directly in the GIS and make new vector layers to create surface flow paths.

  i. *Questions: Describe the flow paths you drew on your map. What was your reasoning for electing to draw the flow paths you did? What challenges or obstacles did you encounter while determining the routes water would take? How did you overcome the challenges/obstacles to*

635   *determine the routes?*

 f. To determine flow paths more objectively, use a software designed with flow-routing algorithms. Here we will use Topographic Analysis Kit (TAK). You first need to give the software the output files name prefix and select the output directory. Then load the Reprojected DEM (tif file) into TAK. If it gives you an error regarding whole numbers, don't worry, your file has still loaded correctly. Check the box labeled

640  "Resample" and then click "Run MakeStreams." This is as far as this activity goes as far with using TAK, but it is a powerful tool for doing geomorphological analysis. If you are curious, I recommend you check out the documentation at Github and explore it further on your own.

 g. From the TAK output file folder, drag and drop your new shapefile into the GIS. If you do not see your streams overlaying the topography, right click on your streams vector layer name. Go to "Properties" >





"General" > "Coordinate Reference System." Select the appropriate CRS. Click on "Update extents", "Apply", "OK."

4.  Data analysis.

a.  Compare the stream network predicted by TAK with the one you drew in step 11. *What similarities or differences do you see between the two networks? Which one do you think is more accurate? Why?*

b.  Using your observations of the geology, the topography, and the hydrology, construct a geologic/geomorphic history of your study area. *What was the sequence of events at this site—consider in particular depositional, tectonic, and erosional events? How did the stream network (or lack thereof) evolve?*

5.  Hypothesis formulation.

a.  Some events in your above history will be more hypothetical than others. Please state which events need additional testing to be able to defend them.

b.  What environmental or natural-disaster hazards do you think might be issues in this landscape? Why do you think this? Write these ideas in the form of additional hypotheses about this landscape.

6.  Experimental design.

a.  What data would you need to collect to support or refute your hypotheses? Please speculate as to the kinds of results that may be obtained for different types of data and what implications those might have for each hypothesis.

b.  What field, laboratory, or numerical techniques would be required to obtain the data you need? Please be specific as if you were planning for field work, lab work, or numerical modeling.

Sharing science:

1.  After completing the exercise, as an individual or as part of a small group, present your findings to the whole class.

2.  Write a formally structured report (Title, author's name, date, abstract, introduction, methods, results, discussion, conclusion). Within the report or as a separate document, reflect on your experience with this activity and assess your level

of understanding before and after the activity of a.) © Google Earth, b.) GIS, c.) UTM CRS, d.) topographic map interpretation, e.) TAK, e.) karst hydrogeology, f.) geologic history construction, g.) hypothesis formulation, and h.) scientific experiment design.





**Appendix C: Karst Geomorphology and Senior Capstone Virtual Learning Experiences Overview in Six Languages**

**C.1 Karst Geomorphology and Senior Capstone Virtual Learning Experiences Overview**

**By Rachel Bosch, karstgeomorph@gmail.com**

Two activities are available on Teaching with Online Field Experiences repository, a page of the Science Education Resource Center (SERC):

- Karst Hydrogeology: A virtual field introduction using © Google Earth and GIS
- Karst Hydrogeology and Geomorphology: A virtual field experience using © Google Earth, GIS, and TAK


This © Google doc contains three Virtual Capstone Pathways for providing a full undergraduate capstone experience.

The following video, From Field to Phone: A Karst Camp Chronicle, was presented on 29 October 202 at the Geological Society of America (GSA) 2020 Connects Online conference. To select subtitles when viewing this video, click on the

settings gear: . A menu will pop up. Click on "Subtitles/CC (7)." Then click on your preferred language to view subtitles with the video.

**Abstract**

During the summer of 2020, many geology field camps were cancelled due to the COVID-19 pandemic, including the Karst

Geomorphology field course I was scheduled to co-teach through Western Kentucky University. When the National Association of Geoscience Teachers in collaboration with the International Association for Geoscience Diversity began the project of supporting working groups to create online field experience teaching material, I saw an opportunity. From the field camp syllabus, I created two products that are now freely available on the SERC Online Field Experiences repository: "Karst Hydrogeology: A virtual field introduction using © Google Earth and GIS" and "Karst Hydrogeology and Geomorphology:

A virtual field experience using © Google Earth, GIS, and TAK (Topographic Analysis Kit)," including student handouts, instructor workflow reference sheets, grading rubrics, and NAGT-established learning objectives. The introductory activity is the more basic of the two, is expected to take about one day to teach, and walks students through all the steps, as well as providing global examples of karst landscapes to virtually explore. The other activity, Karst Hydrogeology and Geomorphology, assumes student familiarity with © Google Earth, GIS, and karst drainage systems, and may take up to two

days to complete. To make these learning opportunities financially accessible, all software required for the activities is open-source and alternative workflows for the introductory module are provided so that the entire exercise can be completed using a smartphone. In addition to providing online capstone activities in the time of a pandemic, these activities provide alternatives to traditional field camps and inclusive learning experiences for all geoscience students. In my home department, I had been contacted by students needing to find capstone experiences when their field camps were cancelled. Responding to

this need and providing a virtual alternative for years to come, I assembled the SERC activities into three learning tracks,





each one providing learning hours equivalent to a traditional field camp, which are being taught at the University of Cincinnati in the Fall Semester of 2020.

**C.2 Descripción general de las experiencias de aprendizaje virtual de geomorfología kárstica y sénior Capstone**

**Por Rachel Bosch, karstgeomorph@gmail.com**

Hay dos actividades disponibles en el repositorio Teaching with Online Field Experiences, una página del Science Education Resource Center (SERC):

- Hidrogeología kárstica: una introducción al campo virtual utilizando © Google Earth y GIS
- Hidrogeología y geomorfología kárstica: una experiencia de campo virtual utilizando © Google Earth, GIS y TAK

Este documento de © Google contiene tres vías virtuales de finalización para proporcionar una experiencia completa de finalización de pregrado.

El siguiente video, From Field to Phone: A Karst Camp Chronicle, se presentó el 29 de octubre de 202 en la conferencia 2020 Connects Online de la Geological Society of America (GSA). Para seleccionar subtítulos al ver este video, haga clic en

el engranaje de configuración.
Aparecerá un menú. Haga clic en "Subtítulos / CC (7)". Luego haga clic en su idioma preferido para ver los subtítulos con el video.

**Resumen**

Durante el verano de 2020, muchos campamentos de campo de geología se cancelaron debido a la pandemia de COVID-19, incluido el curso de campo de geomorfología kárstica que estaba programado para enseñar conjuntamente en la Western Kentucky University. Cuando la Asociación Nacional de Profesores de Geociencias en colaboración con la Asociación Internacional para la Diversidad de las Geociencias comenzó el proyecto de apoyar grupos de trabajo para crear material didáctico de experiencia de campo en línea, vi una oportunidad. A partir del programa del campamento de campo, creé dos

productos que ahora están disponibles gratuitamente en el repositorio de Experiencias de campo en línea de SERC: "Hidrogeología kárstica: una introducción de campo virtual con © Google Earth y GIS" y "Hidrogeología y geomorfología kárstica: una experiencia de campo virtual con © Google Earth, GIS y TAK [Kit de análisis topográfico] ", que incluye folletos para los estudiantes, hojas de referencia del flujo de trabajo del instructor, rúbricas de calificación y objetivos de aprendizaje establecidos por NAGT. La actividad introductoria es la más básica de las dos, se espera que tome

aproximadamente un día para enseñar y guía a los estudiantes a través de todos los pasos, además de proporcionar ejemplos globales de paisajes kársticos para explorar virtualmente. La otra actividad, Hidrogeología y Geomorfología del Karst, asume que los estudiantes están familiarizados con © Google Earth, GIS y los sistemas de drenaje kárstico, y puede tardar



hasta dos días en completarse. Para que estas oportunidades de aprendizaje sean económicamente accesibles, todo el software necesario para las actividades es de código abierto y se proporcionan flujos de trabajo alternativos para el módulo

introductorio para que todo el ejercicio se pueda completar utilizando un teléfono inteligente. Además de proporcionar actividades culminantes en línea en el momento de una pandemia, estas actividades brindan alternativas a los campamentos tradicionales y experiencias de aprendizaje inclusivas para todos los estudiantes de geociencias. En mi departamento de origen, me habían contactado estudiantes que necesitaban encontrar experiencias culminantes cuando se cancelaron sus campamentos. Respondiendo a esta necesidad y brindando una alternativa virtual para los años venideros, reuní las

actividades de SERC en tres vías de aprendizaje, cada una brindando horas de aprendizaje equivalentes a un campamento de campo tradicional, que se imparte en la Universidad de Cincinnati en el semestre de otoño 2020.

### C.3 岩溶地貌和高级 Capstone 虚拟学习经验概述

**瑞秋·博世（Rachel Bosch），karstgeomorph@gmail.com**

"在线教学经验教学"资源库提供了两项活动，这是科学教育资源中心（SERC）的页面：

● 喀斯特水文地质学：使用 © Google Earth 和 GIS 的虚拟领域简介
   ● 喀斯特水文地质和地貌：使用 © Google Earth，GIS 和 TAK 的虚拟现场体验

该 © Google 文档包含三个 Virtual Capstone Pathways，可提供完整的本科生体验。

以下视频（从田野到电话：喀斯特营地纪事）于 202 年 10 月 29 日在美国地质学会（GSA）2020 Connects Online 会议上发表。要在观看此视频时选择字幕，请单击设置齿轮.

将会弹出一个菜单。单击"字幕/ CC（7）"。然后单击您喜欢的语言以查看带有视频的字幕。

### 抽象

在 2020 年夏季，由于 COVID-19 大流行，许多地质野外训练营被取消，包括我计划通过西肯塔基大学共同讲授的喀斯特地貌学野外训练课程。当美国国家地质科学教师协会与国际地球科学多样性协会合作启动支持工作组创建在线实地经验教材的项目时，我看到了一个机会。我从野外训练营的课程表中创建了两种产品，这些产品现在可以在 SERC 在线野外经验存储库中免费使用："喀斯特水文地质：使用 © Google Earth 和 GIS 的虚拟领域介绍"和"喀斯特水文地质和地貌：使用 © Google 的虚拟领域的经验" Earth，GIS 和 TAK [Topographic Analysis Kit]，包括学生讲义

，讲师工作流程参考表，评分标准和 NAGT 制定的学习目标。入门活动是两者中最基础的活动，预计将花费一天的时间进行教学，并引导学生完成所有步骤，并提供可虚拟探索的喀斯特地貌的全球示例。另一项活动是岩溶水文地质和地貌学，假定学生熟悉 © Google Earth，GIS 和岩溶排水系统，并且可能需要两天的时间才能完成。为了使这些学习机会在财务上可访问，活动所需的所有软件都是**开源的**，**并且提供了入**门模块的替代工作流，以便可以使



用智能手机完成整个练习。除了在大流行期间提供在线顶峰活动外，这些活动还为所有自然科学学生提供了传统野
外营地和包容性学习体验的替代选择。为响应这一需求，并提供了未来几年的虚拟替代方案，我将 SERC 活动分为
三个学习途径，每个学习途径的学习时间都相当于传统的野外训练营，而辛辛那提大学将在秋季开学。 2020 年。

**C.4 カルスト地形学とシニアキャップストーン仮想学習体験の概要**

**レイチェルボッシュ、karstgeomorph@gmail.com**
科学教育リソースセンター（SERC）のページである「オンライン実地体験による教育」リポジトリでは、次の 2
つのアクティビティを利用できます。
- 「カルスト水文地質学：© Google Earth と GIS を使用した仮想フィールドの紹介」
- 「カルスト水文地質学と地形学：© Google Earth、GIS、TAK を使用した仮想フィールド体験」

このグーグルドキュメントには、完全な学部生のキャップストーン体験を提供するための 3 つの「仮想キャップ
ストーンパスウェイ」が含まれています。

次のビデオ「フィールドから電話へ：カルストキャンプクロニクル」は、202 年 10 月 29 日に米国地質学会（
GSA）2020 ConnectsOnline カンファレンスで発表されました。このビデオを見ているときに字幕を選択するに
は、設定ギアをクリックします。
メニューがポップアップ表示されます。 「字幕/ CC（7）」をクリックします。次に、ご希望の言語をクリック
して、ビデオの字幕を表示します。

**概要**
2020 年の夏、COVID-19 のパンデミックにより、多くの地質フィールドキャンプがキャンセルされました。これ
には、西ケンタッキー大学で共同指導する予定だったカルスト地形フィールドコースも含まれます。全米地球科
学教師協会が国際地球科学多様性協会と協力して、オンラインのフィールド体験教材を作成するためのワーキン
ググループを支援するプロジェクトを開始したとき、私は機会を見ました。フィールドキャンプのシラバスから
、SERC オンラインフィールドエクスペリエンスリポジトリで無料で入手できる 2 つの製品を作成しました。「
カルスト水文地質学：© Google Earth と GIS を使用した仮想フィールドの紹介」と「カルスト水文地質学と地
形学：© Google を使用した仮想フィールドエクスペリエンス」です。 Earth、GIS、および TAK（地形解析キッ
ト）」。これには、学生の配布物、インストラクターのワークフローリファレンスシート、採点基準、および
NAGT が確立した学習目標が含まれます。入門アクティビティは 2 つのうちのより基本的なもので、教えるのに





約 1 日かかると予想され、すべてのステップを生徒に説明するとともに、仮想的に探索するカルスト地形のグロ
ーバルな例を提供します。もう 1 つのアクティビティであるカルスト水文地質学と地形学は、学生が © Google
Earth、GIS、カルスト排水システムに精通していることを前提としており、完了するまでに最大 2 日かかる場合
があります。これらの学習機会を経済的に利用できるようにするために、アクティビティに必要なすべてのソフ
トウェアはオープンソースであり、入門モジュールの代替ワークフローが提供されているため、スマートフォン
を使用して演習全体を完了することができます。パンデミック時にオンラインの絶頂活動を提供することに加え
て、これらの活動は、すべての地球科学の学生に従来のフィールドキャンプと包括的な学習体験に代わるものを
提供します。私のホーム部門では、フィールドキャンプがキャンセルされたときに絶頂体験を見つける必要があ
る学生から連絡がありました。このニーズに応え、今後数年間の仮想的な代替手段を提供するために、SERC ア
クティビティを 3 つの学習トラックにまとめました。各トラックは、シンシナティ大学で秋学期に教えられてい
る従来のフィールドキャンプと同等の学習時間を提供します。 2020 年。

**C.5 Übersicht über Karstgeomorphologie und Senior Capstone Virtual Learning**

Von Rachel Bosch, **karstgeomorph@gmail.com**

Im Repository "Lehren mit Online-Felderfahrungen", einer Seite des Science Education Resource Center
(SERC), stehen zwei Aktivitäten zur Verfügung:

- Karsthydrogeologie: Eine virtuelle Feldeinführung mit © Google Earth und GIS
- Karsthydrogeologie und Geomorphologie: Eine virtuelle Felderfahrung mit © Google Earth, GIS
und TAK

Dieses © Google-Dokument enthält drei virtuelle Capstone-Pfade, um ein umfassendes Capstone-Erlebnis
für Studenten zu bieten.

Das folgende Video, Vom Feld zum Telefon: Eine Karstlager-Chronik, wurde am 29. Oktober 202 auf der
Konferenz Connects Online der Geological Society of America (GSA) 2020 vorgestellt. Klicken Sie auf
das Einstellungszahnrad, um beim Anzeigen dieses Videos Untertitel auszuwählen. Ein Menü wird
angezeigt. Klicken Sie auf "Untertitel / CC (7)". Klicken Sie dann auf Ihre bevorzugte Sprache, um
Untertitel mit dem Video anzuzeigen.

**Abstrakt**




Im Sommer 2020 wurden viele Geologie-Feldlager aufgrund der COVID-19-Pandemie abgesagt, einschließlich des Karst-Geomorphologie-Feldkurses, den ich an der Western Kentucky University gemeinsam unterrichten sollte. Als die National Association of Geoscience Teachers in Zusammenarbeit
mit der International Association for Geoscience Diversity das Projekt zur Unterstützung von Arbeitsgruppen bei der Erstellung von Online-Lehrmaterialien für Felderfahrungen startete, sah ich eine Gelegenheit. Aus dem Lehrplan des Feldlagers habe ich zwei Produkte erstellt, die jetzt im SERC Online Field Experiences-Repository frei verfügbar sind: „Karsthydrogeologie: Eine virtuelle Feldeinführung mit © Google Earth und GIS " und „Karsthydrogeologie und Geomorphologie: Eine
virtuelle Felderfahrung mit © Google Earth, GIS und TAK [Topographic Analysis Kit] ", einschließlich Handouts für Schüler, Referenzblättern für den Workflow von Lehrern, Bewertungsrubriken und von NAGT festgelegten Lernzielen. Die Einführungsaktivität ist die grundlegendere der beiden Aktivitäten. Der Unterricht wird voraussichtlich etwa einen Tag dauern. Sie führt die Schüler durch alle Schritte und bietet globale Beispiele für Karstlandschaften, die
virtuell erkundet werden können. Die andere Aktivität, Karsthydrogeologie und Geomorphologie, setzt voraus, dass die Schüler mit © Google Earth, GIS und Karstentwässerungssystemen vertraut sind. Die Durchführung kann bis zu zwei Tage dauern. Um diese Lernmöglichkeiten finanziell zugänglich zu machen, ist die gesamte für die Aktivitäten erforderliche Software Open Source und es werden alternative Workflows für das Einführungsmodul bereitgestellt, damit die gesamte Übung mit einem
Smartphone abgeschlossen werden kann. Diese Aktivitäten bieten nicht nur Online-Schlusssteinaktivitäten in Zeiten einer Pandemie, sondern auch Alternativen zu traditionellen Feldlagern und integrative Lernerfahrungen für alle geowissenschaftlichen Studenten. In meiner Heimatabteilung war ich von Studenten kontaktiert worden, die bei der Absage ihrer Feldlager Erfahrungen mit Schlusssteinen sammeln mussten. Als Reaktion auf dieses Bedürfnis und als virtuelle
Alternative für die kommenden Jahre habe ich die SERC-Aktivitäten in drei Lernpfaden zusammengefasst, von denen jeder Lernstunden bietet, die einem traditionellen Feldlager entsprechen, das im Herbstsemester an der Universität von Cincinnati unterrichtet wird 2020.

**C.6 Présentation des expériences d'apprentissage virtuel de la géomorphologie karstique et de la pierre angulaire senior**

**Par Rachel Bosch, karstgeomorph@gmail.com**



Deux activités sont disponibles sur le référentiel «Enseigner avec des expériences de terrain en ligne», une page du Science Education Resource Center (SERC):

- ● Hydrogéologie karstique: une introduction au champ virtuel utilisant © Google Earth et le SIG
- ● Hydrogéologie et géomorphologie karstiques: une expérience virtuelle sur le terrain utilisant © Google Earth, GIS et TAK


Ce document © Google contient trois chemins virtuels pour offrir une expérience complète de fin d'études de premier cycle.

La vidéo suivante, «du terrain au téléphone: une chronique du camp karstique», a été diffusée le 29
octobre 202 lors de la conférence Connects Online 2020 de la Geological Society of America (GSA). Pour sélectionner les légendes lors de la visualisation de cette vidéo, cliquez sur l'engrenage des paramètres. Un menu apparaîtra. Cliquez sur "Sous-titres / CC (7)". Cliquez ensuite sur votre langue préférée pour afficher les sous-titres avec la vidéo.

**Abstrait**
Au cours de l'été 2020, de nombreux camps de géologie de terrain ont été annulés en raison de la pandémie de COVID-19, y compris le cours de terrain de géomorphologie karstique que je devais co-enseigner à l'Université Western Kentucky. Lorsque l'Association nationale des professeurs de géosciences, en collaboration avec l'Association internationale pour la diversité des géosciences, a
lancé le projet de soutien aux groupes de travail pour créer du matériel pédagogique en ligne sur le terrain, j'ai vu une opportunité. À partir du programme du camp de terrain, j'ai créé deux produits qui sont maintenant disponibles gratuitement sur le référentiel en ligne des expériences de terrain du SERC: «l'hydrogéologie karstique: une introduction virtuelle sur le terrain utilisant © Google Earth et le SIG» et «l'hydrogéologie et la géomorphologie karstique: une expérience de terrain
virtuelle utilisant © Google Earth SIG et TAK [Kit d'analyse topographique]», y compris les documents de l'élève, les feuilles de référence du flux de travail de l'instructeur, les rubriques de notation et les objectifs d'apprentissage établis par le NAGT. L'activité d'introduction est la plus élémentaire des deux, elle devrait prendre environ une journée pour enseigner et guider les étudiants à travers toutes les étapes, ainsi que pour fournir des exemples mondiaux de paysages





karstiques à explorer virtuellement. L'autre activité, l'hydrogéologie et la géomorphologie karstiques, suppose que les étudiants sont familiarisés avec © Google Earth, les SIG et les systèmes de drainage karstique, et peut prendre jusqu'à deux jours. Pour rendre ces opportunités d'apprentissage financièrement accessibles, tous les logiciels requis pour les activités sont open-source et des flux de travail alternatifs pour le module d'introduction sont fournis afin que
l'ensemble de l'exercice puisse être effectué à l'aide d'un smartphone. En plus d'offrir des activités de récapitulation en ligne en période de pandémie, ces activités offrent des alternatives aux camps de terrain traditionnels et des expériences d'apprentissage inclusives pour tous les étudiants en géosciences. Dans mon département d'origine, j'avais été contacté par des étudiants qui avaient besoin de trouver des expériences synthétiques lorsque leurs camps sur le terrain avaient
été annulés. Répondant à ce besoin et offrant une alternative virtuelle pour les années à venir, j'ai assemblé les activités SERC en trois pistes d'apprentissage, chacune fournissant des heures d'apprentissage équivalentes à un camp de terrain traditionnel, qui sont enseignées à l'Université de Cincinnati au cours du semestre d'automne 2020.