# Peer review of "Development and Implementation of Virtual Field Teaching Resources: Two Karst Geomorphology Modules and Three Virtual Capstone Pathways"

_Geoscience Communication, 2021_

## Referee Comment (RC2)

**Generally, a referee comment should be structured as follows: an initial paragraph or section evaluating the overall quality of the preprint ("general comments"), followed by a section addressing individual scientific questions/issues ("specific comments"), and by a compact listing of purely technical corrections at the very end ("technical corrections": typing errors, etc.).**

**GENERAL COMMENTS (initial paragraph or section evaluating the overall quality of the preprint)**

The submitted manuscript conveys the efforts towards maintaining field education within the parameters of the pandemic. Its content will be benefiting not only to students that want to learn mor about karst topography/terrain, but also to educators that may want to incorporate these strategies to teach geosciences in a place-based framework. Overall, the manuscript has the potential to become really valuable among the community, but needs major revisions to improve the quality of the article. Please consider the following general comments in the revision of the manuscript:

The terms Karst Camp Chronicle in the title does not provide sufficient information about what is the manuscript. In addition, the term "chronicle" reinforces the idea of using active voice the narrative (as in the author sharing just a personal point of view). Please, consider to expand what "karst" mean in this context and possibly substitute the term "chronicle" for another term. In addition, the majority of the activities are designed to be developed in a computer, not in a phone.

It seems that at the end of each activity, the reader would be benefited with a recommendation section in where the author provides expected limitations that you could be finding while completing this activity.

Section number 6 (Page 20, Lines 343-54) does not seem to be information that is a good fit for this manuscript. Please consider eliminating that section in the revision of the manuscript.

Section 7 (discussion). Have room to be expanded and developed more in detail. Through all the manuscript we can find sentences that can be relocated in this section and discussed in detail.

**SPECIFIC COMMENTS (addressing individual scientific questions/issues)**

Line 29: The term TAK is not introduced in the introductory paragraph (just in the abstract). Please, provide the meaning of the abbreviation TAK.

Line 37: The author describes that students where guided through the technical details of using opensource software. Please, mention what are those technical details and to what opensource software is making reference.

The author describes that part of the activity is to explore a karst landscape. Please, provide more information to contextualize the concept of karst landscape. In addition, please discuss where this "karst landscape" is located, if it is considered to be important in the narrative.

Line 39: the author claims the following: "I considered nonphysical barriers such as limited income or restricted access to other resources". However, the nonphysical barriers discussed are more socio-economic or accessibility barriers. Please, consider to review this statement.

Line 42: Previously (line 39), the author claims that "nonphysical barriers" were considered, but claims that the activities could be completed using a smartphone. Discuss how is the use of a smartphone could not be considered as a nonphysical barrier. In addition, please discuss if other devices (like tablets) could be used and what type of "essentials" are needed (like internet).

Line 52-53: the author claims that "all activities developed as a part of this initiative were expected to address as many of these possible" and followed that sentence with learning objectives. Please discuss what was the targeted goal for the learning objective to cover with the activities.

Line 78: Please describe what are the basic "GIS landscape analysis" that the students will learn by doing this activity.

Line 111-112: Consider to update the following claim" More than 25 percent of the world's population either lives on, or obtains their water from, karst landscapes" with information from the following article: Goldscheider, N., Chen, Z., Auler, A. S., Bakalowicz, M., Broda, S., Drew, D., Hartmann, J., Jiang, G., Moosdorf, N., Stevanovic, Z., & Veni, G. (2020). Global distribution of carbonate rocks and karst water resources. *Hydrogeology Journal*. https://doi.org/10.1007/s10040-020-02139-5

Lines 120-125: At the end of the paragraph the author provides in a summarized way some recommendations to improve the activity. That does not seem to be the appropriate space to do that. Please, consider to relocate that section to another part of the manuscript.

Line 132-138: Please discuss why this examples/location were selected as sites to develop this activity?

Line 252: The author mentions that are "seven discrete choices as in the introductory activity". Please, provide more context for these 7 discrete choices. It is unclear what does it mean discrete in this sentence.

**TECHNICAL CORRECTIONS (compact listing of purely technical corrections at the very end; typing errors, etc.)**

The author constantly uses active voices in the narrative, redirecting the focus to the author experience, instead to report the strategies used to move "from field to phone". Please, consider integrating passive voice in the written narrative.

Figures: It is recommended that all figures are in color.

Line 33: Please, consider to exchange the word "paper" found in the sentence "both activities described in this paper" with the word "manuscript".

Line 120: in order to maintain consistency through the manuscript, consider to share link to download the materials needed to complete the activities. In addition, please consider to dedicate a section in where the reader can find all the websites (e.g., Appendix or to table 1).

Line 228: Previously (line 98) the author mention "searching the internet" as a goal for another activity. Consider substituting "WWW" with "searching the internet" to maintain consistency.

Line 246: change the work "hey" to "they.

Line 258: You do not need to describe what DEM means. That is done previously.

Lines 318-330: this section seems to be out of place. In this paragraph, we observe many anecdotal and reflective examples. Please, consider to construct a section at the end of the article dedicated to this. Move this lines and expand.

---

## Author Comment (AC1)

Dear anonymous referees,

Thank you for the time you took to carefully review my manuscript, originally titled, "From Field to Phone: A Karst Camp Chronicle." Considering referee #2's comments, I have changed the title to, "Development and Implementation of Virtual Field Teaching Resources: Two Karst Geomorphology Modules and Three Virtual Capstone Pathways."

I appreciate the thoughtful comments provided by both of you. I have listed them and given a response to each below. Most of these resulted in changes to the manuscript.

As I was performing revisions, I decided that I would like to eliminate Appendices A and B. Both of these student handouts can be downloaded from their respective activity pages on the SERC website. Those Internet links are provided both in the body of the manuscript and in what is now Appendix C. Would someone please provide their opinion on whether Appendices A and B should remain or be deleted?

Best regards,

Rachel Bosch

| Reviewer | Comment | Response |
|----------|---------|----------|
| Referee #1 | ". . . this is a narrative of the author developing these resources, but there is no discussion of why the lessons were designed the way they were with respect to published work on effective teaching strategies, etc. Similarly, there is no review of prior work on virtual field experiences or discussion of how the presented material related to prior efforts." | The original manuscript did provide some background and context. This has been elaborated upon in the Introduction. |

| | | |
|---|---|---|
| Referee #1 | "Following from that, there is no assessment of these lessons themselves, i.e., were they effective? . . . the next hurdle that raises the bar for publication beyond sharing these materials through sources like NAGT etc, is studying the lessons themselves and evaluating their effectiveness. There are certainly challenges in collecting this data in a rigorous manner, but for something like this to be publishable as a manuscript, this should be attempted." | These materials were recently developed in response to the pandemic, which is in keeping with the theme of this special issue. They have only been deployed once, and for that one implementation, there was a cohort of two students. I interviewed the instructor who taught this, and his comments have been included in the manuscript. This manuscript primarily documents this moment in time of the pandemic, the response to it, the development of these learning tools, and to the extent possible, the implementation of those tools. |
| Referee #1 | "The framework for the capstone project is interesting and potentially valuable, but needs to be expanded and similar to the discussions above, it needs to be considered in the context of existing geoscience education literature (e.g., capstone projects in general, virtual capstone projects or capstone projects which incorporate a virtual component, etc) and there should be some evaluation of how it worked." | The implementation of the karst modules was performed in conjunction with the virtual vapstone pathways so both are now discussed in section 5. |
| Referee #2 | "The terms Karst Camp Chronicle in the title does not provide sufficient information about what is the manuscript. In addition, the term "chronicle" reinforces the idea of using active voice the narrative (as in the author sharing just a personal point of view). Please, consider to expand what "karst" mean in this context and possibly substitute the term "chronicle" for another term. In addition, the majority of the activities are designed to be developed in a computer, not in a phone." | changed title to, "Development and Implementation of Virtual Field Teaching Resources: Two Karst Geomorphology Modules and Three Virtual Capstone Pathways" |
| Referee #2 | "It seems that at the end of each activity, the reader would be benefited with a recommendation section in where the author provides expected limitations that you could be finding while completing this activity." | Thank you for this suggestion. I enjoyed writing those sections (2.5 and 3.5)! |
| Referee #2 | "Section number 6 (Page 20, Lines 343-54) does not seem to be information that is a good fit for this manuscript. Please consider eliminating that section in the revision of the manuscript." | Acknowledged. This section, with its corresponding Appendix of translated abstracts has been deleted. |

| | | |
|---|---|---|
| Referee #2 | "Section 7 (discussion). Have room to be expanded and developed more in detail. Through all the manuscript we can find sentences that can be relocated in this section and discussed in detail." | Discussion has been expanded. |
| Referee #2 | "Line 29: The term TAK is not introduced in the introductory paragraph (just in the abstract). Please, provide the meaning of the abbreviation TAK." | acronyms defined, GIS and TAK |
| Referee #2 | "Line 37: The author describes that students where guided through the technical details of using opensource software. Please, mention what are those technical details and to what opensource software is making reference." | addressed |
| Referee #2 | "The author describes that part of the activity is to explore a karst landscape. Please, provide more information to contextualize the concept of karst landscape. In addition, please discuss where this "karst landscape" is located, if it is considered to be important in the narrative." | karst landscape defined |
| Referee #2 | "Line 39: the author claims the following: "I considered nonphysical barriers such as limited income or restricted access to other resources". However, the nonphysical barriers discussed are more socio-economic or accessibility barriers. Please, consider to review this statement." | Done, thank you. |
| Referee #2 | "Line 42: Previously (line 39), the author claims that "nonphysical barriers" were considered, but claims that the activities could be completed using a smartphone. Discuss how is the use of a smartphone could not be considered as a nonphysical barrier. In addition, please discuss if other devices (like tablets) could be used and what type of "essentials" are needed (like internet)." | edited to include tablets |
| Referee #2 | "Line 52-53: the author claims that "all activities developed as a part of this initiative were expected to address as many of these possible" and followed that sentence with learning objectives. Please discuss what was the targeted goal for the learning objective to cover with the activities." | specific application of objectives listed in Table 1 |
| Referee #2 | "Line 78: Please describe what are the basic "GIS landscape analysis" that the students will learn by doing this activity." | specified the basic GIS analysis skills that are taught in that sentence |

| | | |
|---|---|---|
| Referee #2 | "Line 111-112: Consider to update the following claim" More than 25 percent of the world's population either lives on, or obtains their water from, karst landscapes" with information from the following article: Goldscheider, N., Chen, Z., Auler, A. S., Bakalowicz, M., Broda, S., Drew, D., Hartmann, J., Jiang, G., Moosdorf, N., Stevanovic, Z., & Veni, G. (2020). Global distribution of carbonate rocks and karst water resources. Hydrogeology Journal. https://doi.org/10.1007/s10040-020-02139-5" | good catch--updated |
| Referee #2 | "Lines 120-125: At the end of the paragraph the author provides in a summarized way some recommendations to improve the activity. That does not seem to be the appropriate space to do that. Please, consider to relocate that section to another part of the manuscript." | These are not recommendations to improve the activity. Rather they are options for accessibility. I think that it the appropriate location. |
| Referee #2 | "Line 132-138: Please discuss why this examples/location were selected as sites to develop this activity?" | I added an explanation of selection criteria in the manuscript. |
| Referee #2 | "Line 252: The author mentions that are "seven discrete choices as in the introductory activity". Please, provide more context for these 7 discrete choices. It is unclear what does it mean discrete in this sentence." | I eliminated the work "discrete" to remove possible confusion. Changed wording to "Instead of being provided with seven choices . . . " |
| Referee #2 | "The author constantly uses active voices in the narrative, redirecting the focus to the author experience, instead to report the strategies used to move "from field to phone". Please, consider integrating passive voice in the written narrative." | That is a writing weakness of mine. I tend toward the active voice. I have revised the manuscript in an attempt to maximize the passive and minimize the active. I also extended this exercise to reduce the usage of first-person pronouns. |
| Referee #2 | "Figures: It is recommended that all figures are in color." | They were submitted in color. Not sure why you can't see them in color. |
| Referee #2 | "Line 33: Please, consider to exchange the word "paper" found in the sentence "both activities described in this paper" with the word "manuscript"." | I did a global search on "paper" and changed to "manuscript" in a few other instances, as well. |
| Referee #2 | "Line 120: in order to maintain consistency through the manuscript, consider to share link to download the materials needed to complete the activities. In addition, please consider to dedicate a section in where the reader can find all the websites (e.g., Appendix or to table 1)." | great idea! In response to this, I created Appendix C. |

| | | |
|---|---|---|
| Referee #2 | "Line 228: Previously (line 98) the author mention "searching the internet" as a goal for another activity. Consider substituting "WWW" with "searching the internet" to maintain consistency." | Done, thank you. |
| Referee #2 | "Line 246: change the work "hey" to "they." | Done. Thank you for catching that error. |
| Referee #2 | "Line 258: You do not need to describe what DEM means. That is done previously." | eliminated writing it out in this instance |
| Referee #2 | "Lines 318-330: this section seems to be out of place. In this paragraph, we observe many anecdotal and reflective examples. Please, consider to construct a section at the end of the article dedicated to this. Move this lines and expand." | This has been rewritten more objectively in passive voice. The reflective portion of this paragraph was moved to the Acknowledgements. |